# Textually Pretrained Speech Language Models

**Michael Hassid**$^{\heartsuit,\spadesuit,*}$    **Tal Remez**$^{\spadesuit,*}$    **Tu Anh Nguyen**$^{\spadesuit}$    **Itai Gat**$^{\spadesuit}$

**Alexis Conneau**$^{\diamondsuit}$    **Felix Kreuk**$^{\spadesuit}$    **Jade Copet**$^{\spadesuit}$    **Alexandre Defossez**$^{\spadesuit}$

**Gabriel Synnaeve**$^{\spadesuit}$    **Emmanuel Dupoux**$^{\spadesuit}$    **Roy Schwartz**$^{\heartsuit,*}$    **Yossi Adi**$^{\heartsuit,\spadesuit,*}$

$^{\spadesuit}$FAIR Team, Meta
$^{\diamondsuit}$OpenAI
$^{\heartsuit}$The Hebrew University of Jerusalem

`michael.hassid@mail.huji.ac.il`

## Abstract

Speech language models (SpeechLMs) process and generate acoustic data only, without textual supervision. In this work, we propose TWIST, a method for training SpeechLMs using a *warm-start* from a pretrained *textual* language models. We show using both automatic and human evaluations that TWIST outperforms a cold-start SpeechLM across the board. We empirically analyze the effect of different model design choices such as the speech tokenizer, the pretrained textual model, and the dataset size. We find that model and dataset scale both play an important role in constructing better-performing SpeechLMs. Based on our observations, we present the largest (to the best of our knowledge) SpeechLM both in terms of number of parameters and training data. We additionally introduce two spoken versions of the StoryCloze textual benchmark to further improve model evaluation and advance future research in the field. We make speech samples, code and models publicly available.[2]

## 1 Introduction

*Speech* is the earliest form of human language. Although it contains more than just textual content (e.g., intonation, non-verbal vocalizations), most spoken language understanding systems are limited to its textual form [Wen et al., 2015, Bastianelli et al., 2020, Gupta, 2022]; see Qin et al. [2021] for a recent survey. Recent progress in speech language modeling [Touvron et al., 2023], speech synthesis [Kong et al., 2020], and acoustic unit discovery [Hsu et al., 2021] provided the ability to build purely speech-based language models, henceforth, SpeechLMs [Lakhotia et al., 2021]. However, despite the fast-growing presence of speech and audio content,[3] text is still by far the most dominant language modality on the web. This limits the ability of constructing large SpeechLMs, in contrast to the great success of textual Language Model (LM)s [Devlin et al., 2019, Raffel et al., 2020, Brown et al., 2020, Chowdhery et al., 2022]. The development of large textual LMs trained on massive text corpora allows such models to effectively perform various tasks based on either few examples or textual instructions [Brown et al., 2020, Touvron et al., 2023]. Such LMs often serve as a foundational

---

$^{*}$Core Contributor. $\diamondsuit$ Work done while working at Meta.
$^{2}$https://pages.cs.huji.ac.il/adiyoss-lab/twist/
$^{3}$E.g., podcasts, local radio, and video games. See https://www.insiderintelligence.com/content/look-us-digital-audio-market-2022-how-big-who-s-listening-what-they-listening.

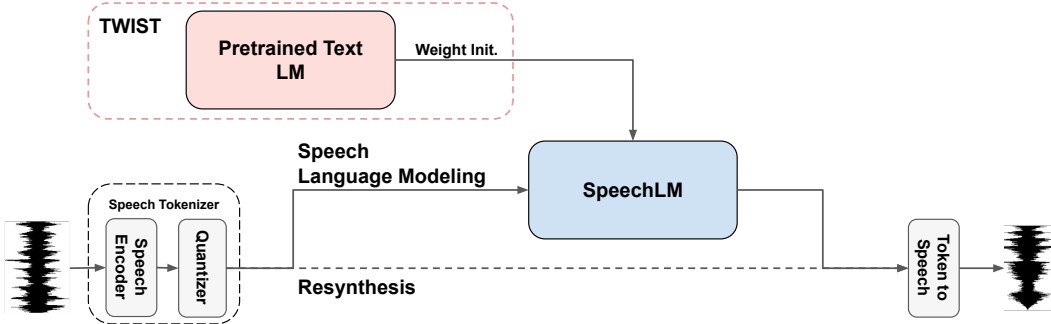

Figure 1: Generative Spoken Language Modeling: the pipeline is composed of three main components (i) Speech tokenizer; (ii) SpeechLM; and (iii) Token-to-speech. This paper introduces TWIST,which initializes the weights of the SpeechLM from a pretrained text LM.

model to be fine-tuned on other tasks such as text classification [Howard and Ruder, 2018], textual instructions [Ouyang et al., 2022], or code generation [Le et al., 2022, Nijkamp et al., 2022].

This success raises the question whether textual LMs can be leveraged to improve SpeechLMs. On the one hand, as both modalities operate on completely different granularity levels (pseudo phoneme-states level vs. sub-word level), it is unclear whether such transfer will bring any benefit. On the other hand, speech and text are closely connected, and thus it is natural to consider transferring models across these modalities. Indeed—previous work was able to train speech and text LMs jointly, focusing on improving speech translation [Bapna et al., 2021, Cheng et al., 2022, Bapna et al., 2022] or transcription tasks [Ao et al., 2021, Chen et al., 2023].

In this work, we show that textual LMs can benefit SpeechLMs. We propose TWIST, **T**extually **W**arm **I**nitialized **S**peech **T**ransformer Language Models, a method for initializing SpeechLMs from a pretrained textual LMs (see Fig. 1). Our empirical findings suggest that such a simple approach is highly effective and provides a consistent improvement across all the evaluated metrics, both automatic and human evaluations. We provide an extensive empirical analysis studying the effect of model and data scale, model architecture, and speech tokenizer on the overall model performance. Based on our observations, we present the largest SpeechLM to date (to the best of our knowledge), both in terms of size (13B parameters) and training data (∼150k speech hours). Finally, to better evaluate SpeechLMs capabilities to model long contextual spoken sentences, we generate two spoken versions of the StoryCloze benchmark [Mostafazadeh et al., 2016]. We synthesize all the stories from the original StoryCloze test set using either the original distractor or a randomly chosen one. Each of the versions evaluates different properties of SpeechLMs; the former captures fine-grained temporal commonsense relation, while the latter captures coarse global sentence coherence.

**Our contributions:** (i) We introduce TWIST, a textually warm initialized speech transformer language model. We empirically show how text-based LMs can be adapted into SpeechLMs, yielding consistent improvements across all evaluated metrics, including both automatic and human ones; (ii) We provide extensive analysis considering the different components of SpeechLMs. Our analysis sheds light on the different model design choices and how they affect model performance; (iii) We leverage our finding and train the largest SpeechLM to-date in terms of the number of parameters and the amount of training data; (iv) We provide two spoken versions of the StoryCloze dataset capturing different aspects of the spoken content. We hope such datasets will be helpful for the research community to better evaluate SpeechLMs under different setups.

## 2 Using Textual LMs to Improve SpeechLMs

We formally describe the proposed method. We start (Section 2.1) by presenting the relevant background, including the Generative Spoken Language Modeling (GSLM) pipeline, which we build on, and relevant LM formulation. We then present TWIST, our proposed method (Section 2.2).

### 2.1 Background

In this work we follow the GSLM framework [Lakhotia et al., 2021]. The general GSLM pipeline is composed of three main modules, each trained separately: (i) a speech tokenizer, (ii) a SpeechLM,

and (iii) a vocoder module (i.e., Token-to-speech). Speech resynthesis can be achieved while ignoring the language model and directly feeding the quantized tokens into the vocoder module [Polyak et al., 2021]. In the following, we provide background for each of the components mentioned above. See Fig. 1 for a visual description.

**Speech tokenizers** encode raw speech into a discrete representation. The common approach is to first encode the speech into a continuous representation and then quantize the representation to achieve a sequence of discrete tokens [Tjandra et al., 2019, 2020, Lakhotia et al., 2021, Borsos et al., 2022].

Formally, denote the domain of audio samples by $\mathcal{X} \subset \mathbb{R}$. The representation for a raw signal is therefore a sequence of samples $x = (x_1, \ldots, x_T)$, where $x_t \in \mathcal{X}$ for all $1 \leq t \leq T$.

Consider an encoder network, $f$, which gets as input the speech utterance and outputs a sequence of spectral representations sampled at a low frequency as follows $f(x) = (v_1, \ldots, v_{T'})$, where $T'$ is determined by the frame rate of the encoder. Note that we do not assume anything about the structure of the encoder network $f$. Lakhotia et al. [2021] evaluated several speech encoders, namely, Mel-spectrogram, Contrastive Predictive Coding [Oord et al., 2018, CPC], wav2vec2 [Baevski et al., 2020], and HuBERT [Hsu et al., 2021]. They found that HuBERT provides the best overall performance, and hence we follow the same setup.

Since the representations learned by such models are usually continuous, a k-means algorithm [MacQueen, 1967] is applied over the models' outputs to generate discrete tokens, denoted as $z = (z_1, \ldots, z_{T'})$. Each element $z_i$ in $z$ is a positive integer, $z_i \in \{1, \ldots, K\}$ for $1 \leq i \leq T'$, where $K$ is the number of discrete tokens of the vocabulary $\mathcal{Z} = \{1, \ldots, K\}$.

**Language models** aim to learn the underlying joint probability of token sequences $p(w_1, \ldots, w_n)$. Each token $w_i$ belongs to a vocabulary $\mathcal{W}$, defined by a tokenizer.[4] Using the chain rule, the joint probability of a sequence can be computed as a product of its conditional probabilities:

$$p(w_1, \ldots, w_n) = \prod_{i=1}^{n} p(w_i | w_{i-1}, \ldots, w_1).$$

Neural LMs, parameterized by $\theta$, aim to model the probability $p_\theta(w_i | c(w_{i-1}, \ldots, w_1))$, where $c$ is a representation of the previous tokens. The network parameters $\theta$ are learned by minimizing the negative log likelihood loss between the predicted and true probability distributions, such that

$$\ell(\theta, w) = -\sum_{i=1}^{n} \log p_\theta(w_i | c(w_{i-1}, \ldots, w_1)).$$

The network parameters $\theta$ are typically initialized with values sampled from a predefined distribution, e.g., a uniform or a centered Gaussian distributions [Glorot and Bengio, 2010].

**Speech Language Models (SpeechLMs)** are trained on the extracted discrete speech tokens, $z$, using a speech tokenizer. When operating on $z$, SpeechLMs enable directly modeling spoken data without accessing textual transcriptions. Such a modeling framework additionally allows for capturing and modeling prosodic features [Kharitonov et al., 2021], as well as speaker identity [Borsos et al., 2022], or even natural dialogues [Nguyen et al., 2022]. This is in contrast to using textual features, which do not encode such information.

**Token-to-speech** modules convert the speech discrete tokens to a raw waveform. Lakhotia et al. [2021] used a Tacotron2.0 [Shen et al., 2018] based model followed by WaveGlow [Prenger et al., 2019] vocoder. Later, Polyak et al. [2021] proposed a token-based vocoder based on the HiFi-GAN architecture to convert tokens to speech directly. Such a paradigm seems to provide high-quality generations with better efficiency, as it uses only one model rather than two. As such, we use their approach in this work for our token-to-speech models.

## 2.2  Textually Warm-Initialized Speech Transformer Language Models

We propose TWIST, a method for training SpeechLMs initialized from a pretrained text LM, such as OPT [Zhang et al., 2022], and LLaMA [Touvron et al., 2023]. TWIST first replaces the original text vocabulary $\mathcal{W}$ with $\mathcal{Z}$, the set of speech tokens (see Section 2.1), and sets the tokenizer to be

---

[4]Typically for text LMs, the tokenizer considers words or sub-words [Sennrich et al., 2016].

Table 1: Speech datasets statistics. We report both the number of hours in the training data of each dataset together with the number of speech tokens used to train the speech language model.

|  | LIBRISPEECH | LIBRILIGHT | SPOTIFY | PEOPLE | VOXPOPULI | OVERALL |
|---|---|---|---|---|---|---|
| Hours | 960 | 53k | 59k | 17k | 20k | 150k |
| Tokens-50Hz | 86M | 4.79B | 5.34B | 1.54B | 1.76B | 13.52B |
| Tokens-25Hz | 58M | 3.19B | 3.56B | 1.02B | 1.17B | 9.00B |

a speech based tokenizer. We then replace the text lookup table with a new randomly initialized embedding table for the speech tokens. The rest of the body of the network remains unchanged during initialization time. Finally, TWIST continues training the entire SpeechLM using speech data. A visual description can be found in Fig. 1.

One might wonder whether initializing a speech model with a textual one makes sense, as speech tokens operate on 20-40ms windows while text tokenizers span longer concepts (e.g., sub-words).[5] Nonetheless, previous work showed that speech and text LMs can be trained jointly to improve either machine translation [Bapna et al., 2021, Cheng et al., 2022, Bapna et al., 2022], or transcription based speech related tasks [Ao et al., 2021, Chen et al., 2023]. We next show that SpeechLMs can benefit from textual LM initialization. We recognize that more advanced methods for converting speech tokens to word tokens probably exist, and hope this study will motivate researchers to explore them.

## 3 Experimental Setup

We follow the same setup as described in Section 2.1. Similarly to Lakhotia et al. [2021], we consider HuBERT [Hsu et al., 2021] followed by a k-means quantizer as the speech tokenizer. We use a token based HiFi-GAN neural vocoder as our Token-to-speech model, with duration predictor as in Polyak et al. [2021], Lee et al. [2021]. Then we optimize, compare and analyze the performance of various SpeechLMs considering different setups.

### 3.1 Datasets

All SpeechLMs are optimized using a collection of publicly available academic speech datasets: LibriSpeech (LS) [Panayotov et al., 2015], LibriLight (LL) [Kahn et al., 2020], Spotify pod-casts [Clifton et al., 2020], People dataset [Galvez et al., 2021], and VoxPopuli [Wang et al., 2021a]. We filter non-English data using the provided meta-data. This results in ∼150k hours, which translate to 13.5B tokens using a token frequency of 50Hz and 9B tokens using a token frequency of 25Hz. See Table 1 for detailed descriptions of the datasets. To the best of our knowledge, this is the first work that uses this scale of data to train SpeechLMs.

In cases where no pre-defined validation and test sets are available, we randomly sample 2% of the data serving as the validation set and an additional 2% for the test set. Unless stated otherwise, in all setups reported results are the average across all different test sets.

### 3.2 Model & Hyperparameters

Our main family of textual LMs is OPT [Zhang et al., 2022]. We examine three model sizes: OPT-125M, OPT-350M and OPT-1.3B, corresponding to 12/24/24 transformer blocks and 768/1024/2048 hidden dimension, respectively.[6] We also consider other pretraining approaches, e.g., different pretraining corpora, and experiment with BLOOM [Scao et al., 2022] and Pythia [Biderman et al., 2023], both with equivalent sizes to OPT-350M/1.3B. For each model size, we compare two variants: TWIST, i.e., a warm initialization from a textual LM, and a cold (randomly initialized) model following the original GSLM approach (henceforth COLD-INIT). We use 8 GPUs for training, except

---

[5]Considering a speech rate of ∼120 words per minute (https://virtualspeech.com/blog/average-speaking-rate-words-per-minute) an average word duration is 500ms, which means that SpeechLMs represent a single word using 12–25 tokens.

[6]The actual number of parameters of the SpeechLMs is about 10–25% lower than the text ones since we replace the text embedding layer with a smaller speech embedding layer, due to the smaller speech vocabulary.

Table 2: Zero-shot modeling results for different number of tokens and downsampling factors (Frequency), with and without TWIST. We report PPL over speech tokens, sWUGGY, sBLIMP. Bold indicates the best model for each tokenizer, and we underline the best performing model overall for sWUGGY and sBLIMP (PPL results are incomparable across tokenizers).

| TWIST | # TOKENS | FREQ. | PPL↓ | sWUGGY↑ | sBLIMP↑ |
|:---:|:---:|:---:|:---:|:---:|:---:|
| ✗ | 100 | 50Hz | 5.26 | 68.91 | 53.80 |
| ✓ | | | **5.03** | **71.30** | **55.96** |
| ✗ | 200 | 50Hz | 5.61 | 69.85 | 53.48 |
| ✓ | | | **5.29** | **72.92** | **55.91** |
| ✗ | 500 | 50Hz | 6.36 | 66.65 | 50.79 |
| ✓ | | | **5.85** | **70.69** | **52.71** |
| ✗ | 500 | 25Hz | 6.65 | 79.44 | 54.84 |
| ✓ | | | **6.25** | **81.42** | **56.20** |

for the 1.3B models, which use 32 GPUs. In all cases, we choose the best checkpoint by the averaged speech perplexity of the model over the validation sets.

For the speech tokenizers, we quantize HuBERT representations, which operate at 50Hz, with 100, 200, and 500 clusters using the k-means algorithm. HuBERT is trained on LS 960h, while k-means are trained over the 'clean-100h' part only from the LS corpus. Models and quantizers are obtained from the textless-lib [Kharitonov et al., 2022]. Inspired by Chung et al. [2021] and Borsos et al. [2022], we also trained a new HuBERT tokenizer with a lower sampling rate. We trained the HuBERT model on a varied mixture of datasets: Multilingual LS [Pratap et al., 2020], Vox Populi [Wang et al., 2021a], Common Voice [Ardila et al., 2019], Spotify [Clifton et al., 2020], and Fisher [Cieri et al., 2004]. More details regarding model setups can be found in Appendix A.1.

## 3.3 Evaluation

Evaluating such a complex pipeline comprised of several different components is a challenging task. We follow Lakhotia et al. [2021], who were the first to propose an evaluation setup for this pipeline. We also introduce a novel evaluation framework based on the StoryCloze textual benchmark [Mostafazadeh et al., 2016]. Overall, we consider three different evaluation setups: (i) Zero-shot modeling; (ii) Human evaluation; and (iii) Spoken StoryCloze.

**Zero-shot modeling metrics.** We use the sWUGGY and sBLIMP [Nguyen et al., 2020] metrics to evaluate lexical and syntactic modeling of the SpeechLMs. In sWUGGY, the network is presented with a pair of utterances, an existing word and a matching non-word, and evaluated on their capacity to attribute a higher probability to the existing word. Unless stated otherwise, we follow the common approach [Lakhotia et al., 2021] and report the "in-vocab" split results. As for sBLIMP, similarly to sWUGGY, the network is presented with a matched pair of speech segments, grammatical and ungrammatical sentences. The task is to decide which of the two is grammatical based on the probability of the sentence. For both measures, we compare the geometric mean over the models' sequence probabilities assigned to each utterance within a pair. We additionally report speech tokens Perplexity (PPL), averaged across all test sets. We note that while PPL is not comparable across different tokenizers, it can be helpful to compare TWIST against COLD-INIT.

**Human evaluation.** To better assess the quality of the generated outputs, we follow the same setup as the Meaningfulness MOS [MMOS; Lakhotia et al., 2021]. In this setup, human raters are presented with several speech continuations (∼10 seconds) following the same speech prompt (∼3 seconds), and are instructed to evaluate how natural (considering grammar, meaning, and diversity) a given sample is, on a scale between 1–5 with increments of 0.5. For speech prompts, we randomly sample a collection of 50 examples from the test sets of three different datasets: LL, LS-clean and LS-other to reach overall 150 samples per evaluated method. We enforce 10 raters per sample and use the CrowdMOS [Ribeiro et al., 2011] package with the recommended recipes for detecting and discarding inaccurate scores.

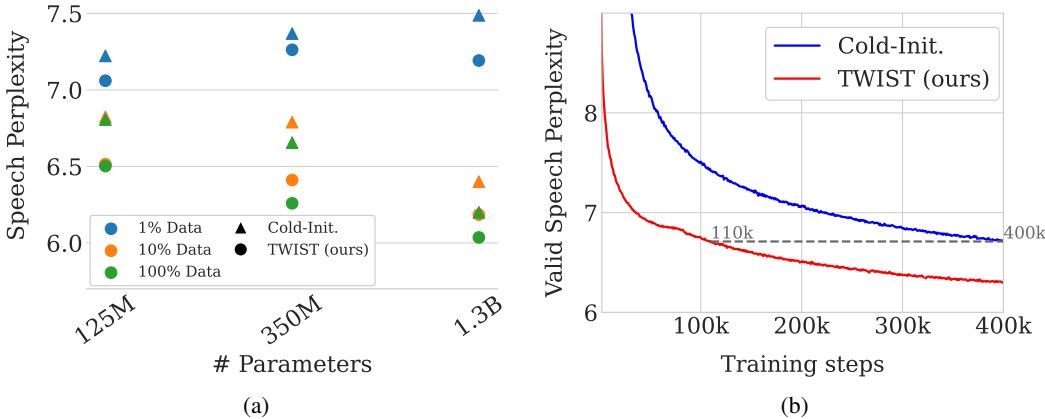

Figure 2: (a) PPL as a function of training set and model sizes. (b) Validation PPL as a function of training steps. TWIST is both more sample-efficient and converges faster than COLD-INIT.

We note that Lakhotia et al. [2021] also proposed computing PPL and auto-BLEU over the transcription of the generated speech to assess both quality and diversity of the generations. While this metrics are automatic, they are highly influenced by numerous factors such as speech vocoder quality, Automatic Speech Recognition (ASR) transcription errors, sampling, etc. In preliminary results, we observe a high variance when computing this metric, hence we focus on a human study, and report these metrics for our main family of models in Appendix A.2.

**Spoken StoryCloze.** Finally, to better evaluate the capabilities of SpeechLMs in capturing fine-grained textual nuances and continuation coherence, we provide two spoken versions of the StoryCloze textual benchmark [Mostafazadeh et al., 2016], denoted by Spoken StoryCloze (SSTORYCLOZE) and Topic StoryClose (TSTORYCLOZE).[7] The textual StoryCloze benchmark contains 4k five-sentence commonsense stories (split to validation and test sets). For each story, there is an additional negative sample, composed of the first four sentences followed by a fifth, adversarial sentence. The goal is to distinguish the original fifth sentence from the negative one. To generate the spoken benchmark, we synthesize the stories from the test set using a single speaker TTS system as provided by Wang et al. [2021b], comprised of a FastSpeech2.0 model [Ren et al., 2020] and a HiFi-GAN vocoder [Kong et al., 2020].[8]

For SSTORYCLOZE, we follow the original StoryCloze negative samples. With this benchmark, we evaluate models' capabilities to capture fine-grained causal and temporal commonsense relations. For TSTORYCLOZE, we randomly sample the negative ending sentence from the dataset. The premise behind TSTORYCLOZE is to evaluate continuation coherence given a spoken prompt. This version is far easier, but our experiments show that it is still quite challenging for modern SpeechLMs. Similar to sWUGGY and sBLIMP, we feed both speech segments to the SpeechLM, measure the probability of each spoken sentence, and report the percentage of examples where the probability of the positive sample is higher than the negative one.

We also conduct human evaluation on both datasets in order to measure human performance over the speech benchmarks (which may differ from their text equivalents). We introduce human raters with both options and let them rate each one according to MMOS [Lakhotia et al., 2021], in a scale of 1–5 with 0.5 increments. The human score for this benchmarks is the proportion of samples where the score given by the rater is higher for the correct utterance. We use 10 raters for each pair of utterances and use the CrowdMOS [Ribeiro et al., 2011] package with the recommended recipes for detecting and discarding inaccurate scores.

---

[7]Both datasets are available at `https://github.com/slp-rl/SpokenStoryCloze`
[8]We introduce a 100ms of silence between segments to generate naturally spoken sentence.

Table 3: Comparison of TWIST-7B/13B models against prior work. We report sWUGGY, using both 'in-vocab' words and 'all' words, along with sBLIMP.

| METHOD | PPL↓ | sWUGGY↑ | | sBLIMP↑ |
|---|---|---|---|---|
| | | all | in-vocab | |
| van Niekerk et al. [2021] | – | 64.3 | 72.3 | 54.0 |
| Lakhotia et al. [2021] | – | – | 68.7 | 57.1 |
| Borsos et al. [2022] | – | 71.5 | 83.7 | **64.7** |
| COLD-INIT-1.3B | 6.20 | 72.2 | 81.9 | 56.5 |
| TWIST-1.3B | 5.93 | 72.7 | 82.5 | 57.0 |
| TWIST-7B | 5.47 | 73.9 | 83.6 | 59.0 |
| TWIST-13B | **5.34** | **74.5** | **84.1** | 59.2 |

## 4 Results

**SpeechLMs benefit from warm initialization using text LMs.** We start by evaluating the effect of the warm initialization on SpeechLMs. We compare two versions of OPT-350M, with warm (using TWIST) and cold initialization (COLD-INIT), for different frequencies and different number of tokens. Results are reported in Table 2. We observe a consistent improvement across all metrics following the TWIST approach. Interestingly, using speech tokens with larger downsampling factor (i.e., smaller frequency) leads to substantially better sWUGGY and sBLIMP results. These findings are consistent with Borsos et al. [2022], and also reflected by the speech resynthesis results, see Appendix A.3. For the rest of the paper we report results using the speech tokenizer with 500 tokens at 25Hz, which gets the best sWUGGY and sBLIMP results. For compatibility with prior work [Lakhotia et al., 2021], results for the speech tokenizer with 200 tokens at 50Hz are in the Appendix.

**Scaling improves SpeechLMs.** As prior work mainly considered relatively small SpeechLMs (e.g., ~100M parameters in Lakhotia et al. [2021], Kharitonov et al. [2021]) and relatively small training data (e.g., Borsos et al. [2022] use LL only), in the following we evaluate the effect of model and dataset scaling on the overall performance.

We train three different SpeechLMs using the OPT family. For each model size, we increase the magnitudes of training data, considering 1%, 10%, and 100% of the training data with both TWIST and COLD-INIT. Our results (Fig. 2a and Fig. 5; Appendix A.4) first highlight that, as in our previous experiments, SpeechLMs initialized using TWIST perform consistently better than those with COLD-INIT. We next note that, as expected, increasing the model size and the magnitude of the dataset improves model performance in almost all cases. Further, for all model scales, following the TWIST approach with only 10% of the data yields comparable or superior performance than the corresponding COLD-INIT approach using 100% of the data. This is consistent with previous findings that pretraining leads to higher sample efficiency for downstream tasks [Peters et al., 2018]. The full set of results for the corresponding models can be found on Table 7 in Appendix A.4.

**TWIST converges faster.** Next, we analyze how textual pretraining affects model convergence. Fig. 2b presents the validation loss training curves for OPT-350M. We observe that using TWIST, the model reaches the same PPL in about one quarter of the training updates compared to COLD-INIT.

**Not all warm initializations are equally important.** We have so far seen that warm initialization from OPT models is consistently beneficial for SpeechLMs. Are OPT models unique in this sense, or could other pre-trained models similarly benefit SpeechLMs?

To address this question, we start by reporting results for different pre-trained text LMs. We consider different textual pre-training approaches—BLOOM [Scao et al., 2022] and Pythia [Biderman et al., 2023]—both of similar size to OPT-350M/1.3B. Tables 8 and 9 in Appendix A.5 shows similar trends to our OPT experiments, in which using textually pretrained LMs yields consistently better results.

We next ask whether initialization of SpeechLMs from pre-trained *text* LMs is particularly important, or would initialization from other pre-trained modalities similarly benefit SpeechLMs. We consider ImageGPT [Chen et al., 2020], an image-generation pre-trained model. We find (Table 10; Appendix A.6) that unlike the textual initialization, this model not only does not outperform COLD-INIT, it substantially *underperforms* it.

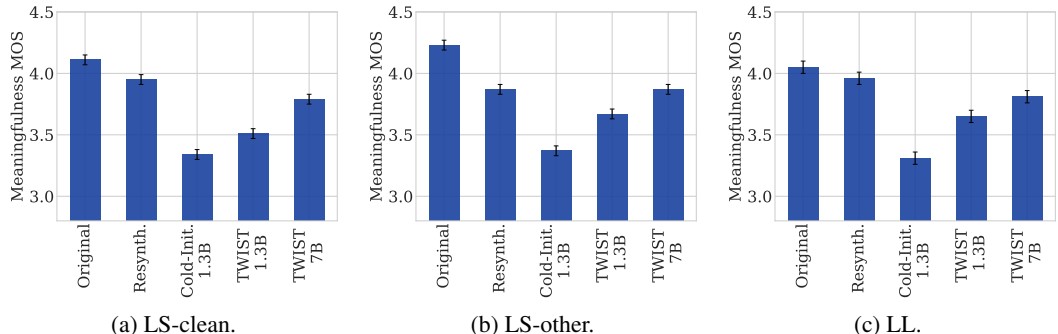

| (a) LS-clean. | (b) LS-other. | (c) LL. |

Figure 3: Human evaluation (MMOS) for speech generation, with different models and datasets. TWIST outperforms the COLD-INIT, and the TWIST-7B performs better than smaller models. Full results presented in Appendix A.7.

**Speech Large Language Models.** Equipped with previous findings, we train the largest SpeechLMs to date (to the best of our knowledge), a 7B/13B parameter SpeechLMs initialized from the LLaMA-7B/13B models [Touvron et al., 2023], denoted as TWIST-7B/13B. We use the same configuration as in Section 3.2 with specific exceptions detailed in Appendix A.1.

Table 3 shows our results for the TWIST-7B/13B models along with several baseline methods. We also include COLD-INIT and TWIST models of size 1.3B for a reference. Here we report sWUGGY scores both for 'in-vocab' samples (words found in the LS corpus) along with 'all' samples (including also words which do not appear in the LS corpus). As expected, scaling benefits SpeechLMs. Initialization from LLaMA-7B/13B leads to additional performance improvement over TWIST-1.3B, ∼8/10% relative improvement in PPL and ∼1.7/2.5% relative improvement in sWUGGY. When comparing to prior work, TWIST-13B outperforms all evaluated methods on both sWUGGY setups.[9] As to sBLIMP, TWIST-13B's results are lower than Borsos et al. [2022], although the proposed method is orthogonal to their method.

**Spoken StoryCloze.** To better asses the contextual understanding of the SpeechLMs we experiment with our collected Spoken StoryCloze benchmark (see Section 3.3).

Table 4 shows the results for TWIST-1.3B/7B/13B models alongside human performance. As expected, both SpeechLMs and humans exhibit superior performance in terms of continuation coherence (TSTORYCLOZE) compared to more fine-grained relations (SSTORYCLOZE). Interestingly, despite humans achieving high scores on the textual StoryCloze benchmark, their performance on spoken benchmarks is not flawless, suggesting that spoken language understanding tasks are more complicated for humans compared to their written equivalents. The results of SpeechLMs indicate

Table 4: Accuracy (%) results for spoken SSTORYCLOZE (SSC) and TSTORYCLOZE (TSC) benchmarks.

| MODEL | SSC↑ | TSC↑ |
|---|---|---|
| Human | 79.9 | 90.2 |
| TWIST-1.3B | 52.4 | 70.6 |
| TWIST-7B | 55.3 | 74.1 |
| TWIST-13B | **55.4** | **76.4** |

reasonable performance on the TSTORYCLOZE benchmark, with a ∼15% gap compared to human performance. However, this gap widens substantially on the SSTORYCLOZE benchmark. This highlights the opportunities for SpeechLMs' improvements both in continuation coherence and, more significantly, in causal and temporal commonsense relations. Lastly, consistent with our previous results, scaling SpeechLMs yields performance improvements on both benchmarks.

**Human evaluation.** We conclude with a human evaluation for speech generation. As stated in Section 3.3, for each model and sample we generate speech continuation of ∼10 seconds (using a ∼3 seconds prompt). We evaluate the following models: 1.3B parameters model with and without TWIST, and TWIST-7B. We also compare to the original audio and the resynthesized reference, which serve as top-line estimates.

Our result (Fig. 3) suggest that TWIST-7B is superior to TWIST-1.3B, while COLD-INIT-1.3B performs consistently worse. Interestingly, we do not observe noticeable differences between the

---

[9]In fact, under the 'all' setup even TWIST-1.3B outperforms all evaluated methods.

```
Prompt: Can we please go swimming at the beach

Cold-Init (1.3B): Can we please go swimming at the beach? Asked Tom. I should
like to go swimming he said...

TWIST-1.3B: Can we please go swimming at the beach? Asked Clara. No, replied
Dora, who had been swimming...

TWIST-7B: Can we please go swimming at the beach with mom on April 1st? Oh my
God, I would love to go somewhere with my kids on April 1st...
```

Figure 4: Generation samples using 1.3B models, with and without TWIST, along with TWIST-7B.

evaluated datasets. We also provide an example of generated sentences given a pre-defined prompt in Fig. 4. This example illustrates that COLD-INIT-1.3B makes grammatical errors, both COLD-INIT-1.3B and TWIST-1.3B remain on topic, and TWIST-7B provides semantically richer continuations. Audio samples can be found in the project website.

## 5 Related Work

Text Language Models have been long studied under various setups [Bahl et al., 1983, Chiang et al., 2022]. While Brants et al. [2007] showed the benefit of optimizing LLMs back in 2007, recently with the help of neural networks such models became foundational models, serving as a base model for different downstream tasks [Shoeybi et al., 2019, Radford et al., 2019, Brown et al., 2020, Lieber et al., 2021, Zhang et al., 2022, Hoffmann et al., 2022, Chowdhery et al., 2022].

The success of pretrained language models as zero-shot or few-shot learners gave rise to an extensive line of work to apply similar technique to other input modalities. Chen et al. [2020] proposed generating natural images while optimizing a language model using masked prediction loss over the pixel space of natural images. Other works proposed converting natural images to discrete tokens space and optimizing textually conditioned language model to perform text-to-image generation [Yu et al., 2022, Ramesh et al., 2021, Chang et al., 2023].

In the context of speech and audio, Lakhotia et al. [2021] first demonstrated how raw and uncurated speech data can be leveraged into building a GSLM system. Next, Kharitonov et al. [2021] proposed a multi-stream SpeechLM to jointly process "pseudo-text" tokens together with quantized prosodic features (i.e., duration and F0). Polyak et al. [2021] evaluated the robustness and disentanglement properties of speech-to-tokens models and demonstrated the ability to perform voice conversion as well as a lightweight speech codec. Kreuk et al. [2021] proposed to cast the task of speech emotion conversion as a translation task, hence translating between one emotion to the other in the discrete space, while Maimon and Adi [2022] proposed a similar approach for speaking style conversion. Nguyen et al. [2022] proposed training two SpeechLMs jointly to mimic natural spoken dialogues. Recently, Borsos et al. [2022] proposed cascading several LMs, in which one LM operates over semantic speech tokens [Sicherman and Adi, 2023] while the others operate on acoustic tokens [Zeghidour et al., 2021, Défossez et al., 2022]. Such modeling framework allows generating natural speech while keeping the identity of the speaker and acoustic conditions unchanged.

Another line of relevant related work, demonstrated that sound and music can be generated following a similar modeling paradigm. Kreuk et al. [2022] first proposed optimizing language models over discrete audio representations to construct a text-to-audio generation model. Similarly Agostinelli et al. [2023] proposed optimizing three language models (following the same modeling approach as by Borsos et al. [2022]), operating at different granularity of the input representation for the task of text-to-music generation. Donahue et al. [2023] proposed a similar modeling approach for the task of singing to accompaniment generation. Lee et al. [2021, 2022], Popuri et al. [2022] followed a similar modeling mechanism using a different speech tokenizer and proposed a textless approach for speech-to-speech translation. Hsu et al. [2022] proposed a jointly modeling visual discrete tokens together with speech tokens to perform various speech resynthesis tasks including: silent video to speech generation, speech enhancement, and recovering packet loss.

Finally, perhaps the line of research most related to this work is jointly training speech and text models. Bapna et al. [2021], Cheng et al. [2022] and Bapna et al. [2022] considered speech as another language in multilingual setups, and showed that involving speech and text as part of training data

improves results for speech translation and multilingual text tasks. Chen et al. [2023] and Ao et al. [2021] used the joint training to improve transcriptions tasks as ASR and Text-to-Speech (TTS). None of these prior work focus on SpeechLMs and its modeling and generation capabilities.

# 6    Conclusion

In this work, we studied the effect of textual pretraining on speech-based language models. We empirically demonstrated how such simple model initialization can improve modeling performance considering both automatic and human-generated measures. We conducted extensive evaluation of different speech tokenizers, model and dataset sizes, model architecture, and modality pretraining. Our analysis sheds light on the impact of specific modeling design choices on the overall performance of the system. Equipped with these findings, we presented the largest SpeechLMs to date (7B and 13B parameters). Finally, we generated two spoken versions of the StoryCloze textual benchmark to measure contextual understanding abilities of SpeechLMs. We hope such empirical findings and benchmark releases will be found useful for future research in the field.

# 7    Limitations and Broader Impact

**Limitations.** The biggest limitation of SpeechLMs at large is the lack of semantic understanding, which might lead to ungrammatical, off-topic, or even inaccurate responses. Although we provided a better initialization point for optimizing SpeechLM, we did not observe major semantic knowledge transfer following the TWIST approach. This limitation is a general critique of SpeechLMs, which should be better studied and addressed in future research. Another limitation of the proposed approach is the granularity of the speech tokenizer, such small input resolution (50Hz / 25Hz) yields a relatively long sequence (750 / 500 tokens) for relatively short speech segments (∼30 seconds). This translates into long inference time and harder optimization.

**Broader impact.** SpeechLMs share the same potential benefits for society as text-based LMs as they give access, in the audio modality, to the same downstream applications (search, language generation, summarization, translation, chatbots, etc.). Thereby increasing their reach to more use cases and more languages, including 'unwritten' (or sparsely written) languages [Lakhotia et al., 2021]. As a result, they also share the same potential risks regarding intentionally harmful applications (e.g. fake news, spamming, election manipulation) or unintentionally harmful ones (e.g., unfair or biased results, toxic, regurgitated or untrustful generations).

# 8    Acknowledgements

We thank Miri Varshavsky Hassid for the great feedback and moral support.

**Authors note.**   This paper was submitted in the wake of the tragic terror attack perpetrated by Hamas on October 7, 2023, which has left the Israeli nation profoundly devastated. The assault greatly impacted us at a personal level, thereby significantly impacting the course of this research work. This paper was finalized while we grieve and mourn our friends and family, under great stress, with scientific considerations being the last thing on our minds. It may contain subtle errors.

In memory of the countless lives shattered by Hamas actions.

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

# A  Supplemental materials

## A.1  Model & hyperparameters

All LM models are trained with a batch size of 64, where each sample is bounded for 25 seconds and 704 tokens. The models are trained for 400k steps (∼1.2 epochs), using an inverse-sqrt scheduler, 100 warmup steps and wADAM as the optimization algorithm. We also tune the learning rate per scenario, i.e: using/not-using pretrained LM, we end up with a maximal learning rate of 4e-4/8e-5 and final learning rate of 8e-5/2.5e-5, respectively. As for the LLaMA-7B/13B model, we use the same configuration except the following: cosine learning rate schedule, 500 warmup steps, a maximum learning rate of 1e-4, a final rate of 1e-5, batch size of 1024 over 32 GPUs for 75k steps (∼4 epochs).

The new the frequency HuBERT speech tokenizer (now available in textless-lib Kharitonov et al. [2022]), is trained for 3 iterations with the default 50Hz features rate. For the 4-th iteration, we add an additional convolutional layer at the CNN Encoder with the strides 2/3/4, resulting in features of 25Hz/16.6Hz/12.5Hz, respectively. Our early ablations show that 25Hz features with 500 tokens give the best results in terms of language modeling, we thus train our models on these new tokens and compare them with the rest of the tokens.

## A.2  Automatic evaluation of speech generation

As stated, PPL scores over the transcribed speech generations (text-PPL) are sensitive to modification in the sampling parameter, vocoder quality and ASR errors [Lakhotia et al., 2021]. Furthermore, text-PPL tells only one side of the story, we also need to count for diversity in the generation, as suggested by Lakhotia et al. [2021]. For instance, repeated speech gets good text-PPL score, at the cost of bad diversity measures.

To mitigate that, when computing the automatic metrics for generated speech, we threshold the temperature parameter using the ASR model confidence. Next, as there can be trade-offs in terms of sentence coherence and diversity, we calibrated the temperature so that the generated speech matches the transcription of the original utterance (as much as possible) in terms of generation diversity (auto-BLEU from Lakhotia et al. [2021]).

For speech prompts, we randomly sample a collection of 1024 examples from the test sets of three different datasets: LS-clean, LS-other and LL. Each speech prompt is ∼3 seconds long, while the continuation is ∼10 seconds. For the calibration procedure, we used randomly selected 128 examples from the corresponding validation datasets. Text transcriptions extracted using Whisper "small" [Radford et al., 2022], while text-PPL measures were computed using a LLaMA-7B (text model) [Touvron et al., 2023]. Table 5 reports both text-PPL and auto-BLEU results for the original continuations, and the generations that produced by our main family of models (OPT architecture coupled with the 25Hz tokenizer).

Table 5: text-PPL and auto-BLEU results, with and without TWIST.

| TWIST | # PARAM. | TEXT-PPL↓ | AUTO-BLEU↓ |
|:---:|:---:|:---:|:---:|
| — | Original Cont. | 36.35 | 0.23 |
| ✗ | 125M | 88.17 | 0.305 |
| ✓ | | **74.95** | **0.29** |
| ✗ | 350M | 117.17 | 0.27 |
| ✓ | | **92.69** | **0.25** |
| ✗ | 1.3B | 67.68 | **0.29** |
| ✓ | | **47.8** | **0.29** |

As can be seen, while the diversity metric is comparable across models (due to calibration), TWIST models outperform the COLD-INIT ones in terms of text-PPL. These measures further support our claims: TWIST models outperform COLD-INIT ones, and models generally improve with scale. Notice, the text-PPL for the 350M models are worse than the 125M models. However, as stated, the diversity of the generations (as computed using auto-BLEU) is better for the 350M models.

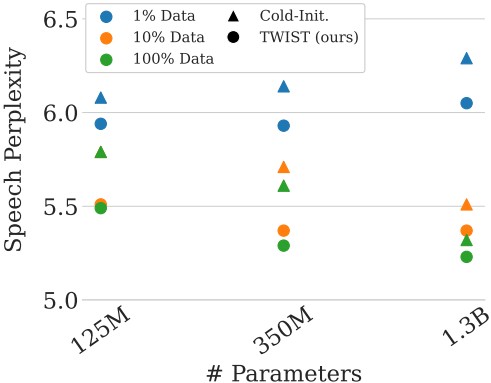

Figure 5: PPL as a function of training set and model size, for models trained with the 200 tokens at 50Hz tokenizer.

## A.3 Speech resynthesis results

Resynthesis can be considered as an upper bound for our language modeling setup. It does not involve SpeechLMs, and measures our ability to fully recover the speech content after tokenization [Polyak et al., 2021]. As we additionally evaluate several speech tokenizers, we provide resynthesis metrics in the form of Word Error Rate (WER). We use Whisper "small" [Radford et al., 2022] as our ASR model.

In Table 6, we evaluate the effect of the tokenizer on the resynthesis performance, and can better evaluate the impact of the tokenization process on the generated audio. As can be seen, all tokenizers incur a loss in WER. Using 500 tokens at 25Hz provides the best performance.

Table 6: Speech Resynthesis. Results are reported for different number of tokens and downsampling factors (Frequency).

| # TOKENS | FREQUENCY | WER↓ |
|---|---|---|
| 100 | 50Hz | 0.23 |
| 200 | 50Hz | 0.18 |
| 500 | 50Hz | 0.17 |
| 500 | 25Hz | **0.16** |

## A.4 Model and data scaling results

The full set of results, i.e., PPL, sWUGGY and sBLIMP from Section 4 for model and dataset scaling is presented in Table 7. The equivalent of Fig. 2a using 200 tokens at 50Hz tokenizer can be found in Fig. 5.

## A.5 The effect of LM architecture

To further validate our findings holds for other LM architectures other than OPT. In Table 8, we provide results for the BLOOM [Scao et al., 2022] and Pythia [Biderman et al., 2023] of similar size to OPT-350M, with both 25Hz and 50Hz tokenizers. In Table 9 we provide results for BLOOM and Pythia of similar size to OPT-1.3B with the 25Hz tokenizer.

As before, we observe similar patterns in terms of using a pretrained text LM. SpeechLMs initialize from text reach better performance across all metrics.

## A.6 The effect of different modality pretraining

Although having completely different granularity, results suggest training SpeechLMs with model initialization from a text based LMs brings a consistent performance improvement. As a result, a

Table 7: Model and Data Scaling. Results are reported for different models on various size using different magnitude of data, with and without TWIST. We report PPL/ sWUGGY / sBLIMP.

| TWIST | # PARAM. | # TOKENS | FREQ. | 1% OF DATA | 10% OF DATA | 100% OF DATA |
|---|---|---|---|---|---|---|
| ✗ | 125M | 200 | 50Hz | 6.08 / 66.90 / 52.45 | 5.79 / 68.16 / 52.71 | 5.79 / 68.26 / 53.02 |
| ✓ | | | | **5.94 / 69.48 / 52.87** | **5.51 / 70.67 / 54.34** | **5.49 / 70.75 / 53.92** |
| ✗ | 350M | 200 | 50Hz | 6.14 / 66.49 / 51.97 | 5.71 / 68.85 / 53.13 | 5.61 / 68.95 / 53.48 |
| ✓ | | | | 5.93 / **68.49 / 53.13** | **5.37 / 72.64 / 55.63** | **5.29 / 72.92 / 55.91** |
| ✗ | 1.3B | 200 | 50Hz | 6.29 / 64.91 / 52.18 | 5.51 / 71.39 / 54.58 | 5.32 / 72.83 / 55.12 |
| ✓ | | | | **6.05 / 67.89 / 52.83** | **5.37 / 72.45 / 55.65** | **5.23 / 73.39 / 55.91** |
| ✗ | 125M | 500 | 25Hz | 7.22 / 77.58 / 53.74 | 6.82 / 78.12 / 54.00 | 6.81 / 77.74 / 54.27 |
| ✓ | | | | **7.06 / 78.99 / 54.12** | **6.52 / 80.08 / 55.45** | **6.50 / 80.57 / 55.43** |
| ✗ | 350M | 500 | 25Hz | 7.37 / 76.96 / 53.07 | 6.79 / 77.93 / 54.71 | 6.65 / 79.44 / 54.84 |
| ✓ | | | | **7.26 / 78.67 / 53.95** | **6.41 / 81.23 / 56.08** | **6.26 / 79.44 / 56.20** |
| ✗ | 1.3B | 500 | 25Hz | 7.49 / 75.44 / 52.96 | 6.40 / 80.80 / 55.95 | 6.20 / 81.94 / 56.52 |
| ✓ | | | | **7.19 / 78.00 / 53.90** | **6.19 / 82.28 / 56.81** | **5.93 / 82.49 / 57.05** |

Table 8: LM Model Architecture. Results are reported for both Bloom and Pythia model architectures (∼350M parameters), with and without TWIST. We report PPL, sWUGGY and sBLIMP.

| TWIST | ARCH. | # TOKENS | FREQ. | PPL↓ | sWUGGY↑ | sBLIMP↑ |
|---|---|---|---|---|---|---|
| ✗ | BLOOM | 200 | 50Hz | 5.63 | 69.38 | 53.03 |
| ✓ | | | | **5.21** | **72.48** | **55.79** |
| ✗ | BLOOM | 500 | 25Hz | 6.45 | 79.46 | 55.60 |
| ✓ | | | | **6.06** | **82.01** | **57.22** |
| ✗ | Pythia | 200 | 50Hz | 5.62 | 70.00 | 53.07 |
| ✓ | | | | **5.23** | **72.20** | **56.00** |
| ✗ | Pythia | 500 | 25Hz | 6.45 | 79.82 | 55.45 |
| ✓ | | | | **6.12** | **81.41** | **56.70** |

Table 9: LM Model Architecture. Results are reported for both Bloom and Pythia model architectures (∼1.3B parameters), with and without TWIST. We report PPL, sWUGGY and sBLIMP.

| TWIST | ARCH. | PPL↓ | sWUGGY↑ | sBLIMP↑ |
|---|---|---|---|---|
| ✗ | BLOOM | 6.09 | 80.47 | 56.02 |
| ✓ | | **5.80** | **82.63** | **57.43** |
| ✗ | Pythia | 6.05 | 81.33 | 56.34 |
| ✓ | | **5.81** | **81.77** | **57.02** |

natural question would be *do speech and text tokens have special connection or LMs are just general next token prediction mechanisms?*

To evaluate such a hypothesis, we consider a language model trained on a different modality. Specifically, we train ImageGPT [Chen et al., 2020] ("medium" size) models, one from scratch and another one pretrained using next pixel prediction using a transformer language model. For the pretrained model we use the official pre-trained model.[10] Table 10 summarizes the results.

Interestingly, ImageGPT pre-trained models perform much worse than models pretrained on text. For a reference, models trained from scratch achieve comparable performance to previously reported models.

---

[10] https://huggingface.co/docs/transformers/model_doc/imagegpt

Table 10: Results for the ImageGPT model (image pretraining), with and without TWIST. We report PPL, sWUGGY and sBLIMP. Unlike textual pretraining, image pretraining not only does not benefit SpeechLMs, but substantially hurts their performance.

| TWIST | # TOKENS | FREQ. | PPL↓ | sWUGGY↑ | sBLIMP↑ |
|---|---|---|---|---|---|
| ✗ | 200 | 50Hz | **5.22** | **72.22** | **55.70** |
| ✓ | | | 8.21 | 54.26 | 51.90 |
| ✗ | 500 | 25Hz | **6.20** | **81.85** | **56.39** |
| ✓ | | | 7.85 | 72.50 | 51.84 |

## A.7 Full human evaluation (MMOS) results

We include the full Human evaluation (MMOS) results, corresponds to Fig. 3 in Table 11.

Table 11: Full human evaluation (MMOS) results. We report MMOS score as: mean (95% confidence-interval).

| METHOD | LS-CLEAN | LS-OTHER | LL-OTHER |
|---|---|---|---|
| Original | 4.11(±0.04) | 4.23(±0.04) | 4.05(±0.05) |
| Resynthesis | 3.95(±0.07) | 3.87(±0.06) | 3.96(±0.06) |
| COLD-INIT-1.3B | 3.34(±0.08) | 3.37(±0.06) | 3.31(±0.07) |
| TWIST-1.3B | 3.51(±0.07) | 3.67(±0.07) | 3.65(±0.06) |
| TWIST-7B | **3.79(±0.06)** | **3.85(±0.07)** | **3.81(±0.06)** |

