# OpenReview forum: "Textually Pretrained Speech Language Models"
_NeurIPS.cc/2023/Conference — NeurIPS 2023 poster_

### Official Review · Reviewer_5UoA · 2023-07-04

**Soundness:** 3 good
**Presentation:** 2 fair
**Contribution:** 2 fair
**Rating:** 6
**Confidence:** 4

**Summary:**

The authors explain a method to improve the performance of a speech language model by reusing the weights of a language model trained on text.

**Strengths:**

Using a well-trained text-based language model as an initial model seems like a good idea.
If the authors' claims can be generalized, it can be used as a good starting point for speech language models, which are relatively difficult to train.

**Weaknesses:**

The authors empirically discovered that they achieved better performance in the same training step, but there seems to be a lack of theoretical analysis. The process of analyzing, estimating, and confirming evidence for why such improvement occurs is missing. This simple empirical discovery without the analysis and validation of why it happens would have limited contribution to the conference-level community.

**Questions:**

I would like to ask the authors for additional comments regarding the following points and the weaknesses mentioned.

The audio samples provided by the authors are not sufficient. While the authors support their claims through various experiments in the manuscript, only some of the samples are included. While source code is the most clear evidence for the experiments and results, it is necessary to present other detailed experimental results when the source code is not available. Furthermore, in the field of speech synthesis, subjective evaluations hold more weight than objective evaluations, so it is essential to provide reviewers with the opportunity to listen to and evaluate the samples directly.

The method proposed by the authors is a form of transfer learning. However, when observing the training phase's perplexity (PPL) curve presented by the authors, it is difficult to consider the model as having converged at the 400k step where the authors claim to have ended the training. Therefore, it is not clear whether the performance difference claimed by the authors is simply due to the difference in convergence time caused by the initialization of the initial weights or the advantages obtained from the text-based language model (such as having trained on larger data, more diverse contexts, etc.).

The MMOS results are presented in a table. It is difficult to verify the confidence intervals (CI). If the CI cannot be verified, it should be indicated how many evaluators participated and with how many samples.


**Limitations:**

The limitations have been well described.

---

> ### Author Rebuttal · Authors · 2023-08-08
>
> We thank the reviewer for acknowledging that using text LMs as initialization for SpeechLMs is a good idea, as our empirical results demonstrate.
>
> **Regarding theoretical justification:** We agree that providing theoretical justification of why TWIST outperforms Cold-Init models is interesting. However, such theoretical justification is far from trivial. Furthermore, offering techniques (such as TWIST) that result in consistent and substantial performance improvements, even in the absence of a well-defined theoretical rationale, holds considerable potential to make a meaningful and beneficial contribution to the broader community, for instance, see [1].
>
> We hypothesize the benefit of TWIST comes from providing a better prior initialization of the model weights for building SpeechLMs. This allows better capturing of long range dependencies in the input sequence (something that we do not observe in image data for instance. See Table 8 in the Appendix). In the attached rebuttal pdf file (Figure.1), we visualize the performance gap between TWIST and Cold-Init. model (350M params), using tStoryCloze, while slowly decreasing the context length. We observe that as the bigger the context, the bigger the gap between the models. We believe such findings will be valuable and of high-interest to the community, especially at the intersection between written and spoken language.
>
> **Regarding audio samples and source-code:** In the appendix material, we provide 15 samples from the proposed method. Per the reviewer's request, we provide an additional 450 samples (link was sent to the AC as per the NeurIPS guidelines), those samples were sent to the MMOS evaluation. As for source-code, we will open source both code and pretrained models.
> However, we would like to emphasize that our contribution in this work is not synthesis quality but improved spoken language modeling (i.e., improving what is said rather than how it sounds). Improving speech generation is orthogonal to our method, and future research could improve generation quality on top of TWIST.
>
> **Regarding model convergence:** We agree that if we had infinite compute and infinite data both models might converge to the same point. However, we note that we used the biggest publicly available speech data so far for training speechLMs (that we are aware of), so scaling further is far from trivial, and further, our setup is already quite expensive as it is (training the 1.3B model for 400k steps takes ~9 days on 32 GPUs). Hence, for fair comparison, we decided to limit each run to 400k steps (which is reasonable considering the change in the training loss). Under this setup we do observe better performance across all settings and faster convergence as can be seen in Figure 2a.
> Furthermore, we do train our LLaMA model until convergence, the graphs are presented in the PDF attached to the rebuttal (Figure 2). One can see that, even at full convergence, the TWIST model outperforms the Cold-Init model in terms of speech PPL (similarly to Figure 2b).
>
> **Regarding source of performance difference:** we believe the performance improvement of TWIST is due to both better initialization and the data used to initialize the model: (i) for better initialization: we do observe the model gets better loss values and overall scores (sWUGGY, sBLIMP, etc.) using TWIST from the beginning of the training, hence we conclude TWIST provide better prior weight initialization; (ii) for the type of data used for initialization: we do not observe performance boost when we use ImageGPT (using image data) as initialization for TWIST, which suggests that the type of data also plays an important role in the performance improvement obtained by TWIST.
>
> **Regarding CI and other details for MMOS:** we include the full table of all MMOS scores used for Figure 3 (we will include these results in the manuscript). As for details regarding the number of raters and number of examples, please see line 175, section 3.3 and line 267in section 4, human evaluation paragraph: we use 50 samples per dataset (total of 150 samples), while we enforce 10 raters per sample.
>
> |     Method    |   Mean   | CI@95 |   Mean   | CI@95 | Mean | CI@95 |
> |:-------------:|:--------:|:-----:|:--------:|:-----:|:----:|:-----:|
> |               | LS-clean |       | LS-other |       |  LL  |       |
> | Ref.          |   4.11   |  0.04 |   4.23   |  0.04 | 4.05 |  0.05 |
> | Resynth       |   3.95   |  0.07 |   3.87   |  0.06 | 3.96 |  0.06 |
> | no-TWIST 1.3B |   3.34   |  0.08 |   3.37   |  0.06 | 3.31 |  0.07 |
> | TWIST 1.3B    |   3.51   |  0.07 |   3.67   |  0.07 | 3.65 |  0.06 |
> | TWIST 7B      |   3.79   |  0.06 |   3.85   |  0.07 | 3.81 |  0.06 |
>
>
>
> [1] Peters, Matthew E., et al. "Deep contextualized word representations." Proceedings of NAACL-HLT. 2018.

---

> > ### Comment · Reviewer_5UoA · 2023-08-15
> >
> > I appreciate the detailed explanations.
> > I have better understood the authors' work through the rebuttal and attached audio samples. I agree that the base model trained with large-scale text data would have improved performance, and the proposed method showed good results. However, considering the level of the conference, supplementation of academic analysis and evidence can be needed beyond empirical things.
> > I agree with most authors' opinions and will update my score to 6.

---

> > > ### Author Response · Authors · 2023-08-16
> > >
> > > We greatly appreciate your insightful feedback. We're delighted that our rebuttal and the additional audio samples proved helpful, and thankful for your input and the revised score.

---

### Official Review · Reviewer_pWSQ · 2023-07-06

**Soundness:** 3 good
**Presentation:** 3 good
**Contribution:** 3 good
**Rating:** 6
**Confidence:** 4

**Summary:**

This work proposes TWIST, Textually Warm-Initialized Speech Transformer-based LMs, a technique to initialize SpeechLMs with pretrained textual LMs. Different textual LMs, tokenizers, models and dataset sizes are evaluated using TWIST. The authors find that a warm start with a textual LM helps compared to a random initialization when evaluated on a number of metrics.

**Strengths:**

- TWIST shows how a warm start with any pretrained textual LM (OPT, Bloom, Pythia) benefits a speech-based LM.
- This is one of the first works to use large-scale audio data (i.e., approximately 150K hours of speech) to train a speech-based LM.
- This work presents a new evaluation benchmark for speech-based LMs based on StoryCloze.

**Weaknesses:**

Warm start with pretrained textual LMs was found to be effective for speechLMs. A more detailed analysis of the learned representations with and without TWIST would have been useful for the reader. I elaborate on this further in my questions below.

**Questions:**

- An analysis of the learned speech-based discrete tokens with and without the use of TWIST would be interesting to see. For example:
    - Visualizing the representations learned via TWIST (i.e., a warm initialization from a textual LM) and those learned via a randomly initialized model. Color-coding representations based on phoneme labels (derived from a forced alignment) might reveal that phoneme distinctions are less pronounced with using TWIST.
    - Training simple probing classifiers (e.g., to recognize phones or subword textual units) whose inputs are speech representations with and without TWIST and comparing their accuracies.
- Due to the warm start from textual LMs that model long-term textual dependencies, one would assume that TWIST would do well on measures like sBLIMP that look at the grammaticality of entire sentences. However, from Table 3, TWIST does not fare as well on sBLIMP as prior work from Borsos et al. Could the authors comment on this?  Also, it would be useful to show numbers using a randomly initialized model (in direct comparison to TWIST) in Table 3.
- Are the results in Table 2 using TWIST with OPT-350M or OPT-1.3B?  Please clarify. Assuming it is 1.3B, why is the sBLIMP score for TWIST-1.3B in Table 3 (60.3) different from that reported in Table 2 (59.3)?

**Limitations:**

Limitations have been addressed in the submission.

---

> ### Author Rebuttal · Authors · 2023-08-08
>
> We thank the reviewer for acknowledging the extensive evaluation we conducted across different textual LMs, tokenizers, models and dataset sizes. We also appreciate the reviewer for noting that our work is one of the first ones to work on large-scale speech language modeling and the introduction of new benchmarks for speech language modeling.
>
> **Regarding analysis of the proposed method:** We thank the reviewer for their suggestion, however, we would like to emphasize that both cold- and warm-started models use the same tokenizer (obtained by a HuBERT model). Once analyzing such discrete representation, we already observe a strong correlation to phoneme states (even without any SpeechLM optimization involved). We can consider such input to the model as “pseudo-phonemes”. Similar findings can also be observed in prior work (see [1]). Hence, we do not expect to see strong trends in better-capturing phoneme information between TWIST vs. non-TWIST models.
>
> We hypothesize the performance improvement we observe while using TWIST is due to better prior weight initialization, which provides better modeling of long sequences. Following the reviewer’s request, we visualize the performance gap between TWIST and non-TWIST model (350M params) while slowly decreasing the context length (see Figure.1 in the attached rebuttal pdf file). We observe that as the bigger the context the bigger the gap between the models.
>
> We additionally provide below a few textual examples from tStoryCloze in which the TWIST model succeeds in modeling, while the non-TWIST one does not. \
> Each example is followed with the correct / incorrect suffix.
>
> *Example 123:*
>
> *Harry went to the theme park with his family. His dad and brother rode on a big roller coaster with him. He then rode on some smaller rides with his mom. The family ate at a restaurant at the park. \
> Harry had a great time at the park. / After counting the cars, we went back home.*
>
> *Example 223:*
>
> *Jennifer forgot to close the front door when she got home. By the time she noticed her pet cat had disappeared. She walked around the neighborhood calling for the cat. She made flyers with her contact information. \
> Jennifer's cat came home the next day, acting very hungry. / She decided to watch her favorite comedy show on Netflix.*
>
> *Example 456:*
>
> *The sea was frightening during a storm. Marcus hated the way the boat rose up and crashed down. It made him sick to his stomach. He cowered in his cabin and shut his eyes but it didn't help. \
> Marcus then vomited from sea sickness. / The worker thanked Mark for his patronage.*
>
> As can be seen, the provided examples consistently demonstrate that the accurate suffix incorporates a name that offers a clue to the correctness of the suffix itself. Notice that the last appearance of the name in the prefix is positioned at a considerable distance from the suffix, (initial or second sentences of the prefix). We note that this property does not hold for all examples. This provides further evidence that TWIST models can capture long range dependencies better.
>
> **Regarding sBLIMP scores:** The AudioLM (Borsos et al.) model consists of a cascade of three Transformer models and uses a different speech tokenizer (which is not publicly available). We hypothesize the different tokenizer is the main factor that affects the sBLIMP results, which is why we do a systematic comparison using the same tokenizer and same model, with and without TWIST throughout the paper (e.g., Tables 2 & 6). We also highlight that our contribution is orthogonal to that of AudioLM, which could additionally use TWIST and potentially get a further performance improvement.
>
> **Regarding adding a randomly initialized model for Table 3:** Thank you for your comment, we will add that result to Table 3.
>
> **Regarding results in Table 2:** We use the OPT-350M, we will clarify.
>
>
> [1] Sicherman, Amitay, and Yossi Adi. "Analysing discrete self supervised speech representation for spoken language modeling." ICASSP 2023-2023 IEEE International Conference on Acoustics, Speech and Signal Processing (ICASSP). IEEE, 2023.

---

> > ### Author Response · Authors · 2023-08-19
> >
> > Dear Reviewer pWSQ,
> >
> > We would like to thank you once again for taking the time to review our manuscript! Your comments and feedback are highly appreciated.
> >
> > We would like to know whether our rebuttals addressed your previous comments. If they did, we would greatly value an increase of the score.
> > Please, let us know if you have any follow-up concerns or comments. We would be happy to clarify those.

---

> > > ### Comment · Reviewer_pWSQ · 2023-08-22
> > >
> > > Thanks to the authors for their response. After having read the other reviews and rebuttals as well, I will raise my score to 6.

---

### Official Review · Reviewer_6BNW · 2023-07-08

**Soundness:** 3 good
**Presentation:** 3 good
**Contribution:** 3 good
**Rating:** 5
**Confidence:** 4

**Summary:**

This paper proposes a simple initialization method for speech LMs named, TWIST (Textually Warm-Initialized Speech Transformer Language Models). Instead of cold-initializing a speech LM, twist initializes it with LLM weights (minus the token embeddings, which are replaced with speech vocabulary). The authors show than on OPT models of size 125M to 1.3B (+ BLOOM, Pythia of 350M size), the proposed method of initialization helps in convergence and final perplexity evaluation, as well as shows benefits over cold initialization over human evaluation (MMOS) of generated completions. The authors also create two spoken versions of the StoryCloze dataset.

**Strengths:**

1. The proposed method is quite simple (just a straight-forward initialization trick) and the empirical results show small benefits across speech LMs of different sizes.

**Weaknesses:**

1. The experiments could be more comprehensive, e.g., only one size of BLOOM/Pythia is considered. Similarly, the authors do not report the  PPL over the transcription of the generated speech. Even if the results are high-variance, it is an important aspect of evaluation as far as the quality and diversity of speech continuations is considered.
2. The proposed method is extremely simple, it just initializes the speech LM with LLM weights. There is prior empirical justification to do this, however, the weight spaces as quite incongruent, as acknowledged by the authors, and this problem hasn't been addressed at all. As such, it is hard to consider the contribution very solid in terms of its technical merit. The authors show that text based initialization outperforms Image based initialization, but that again is not a very novel result.

**Questions:**

1. Why not include the PPL over speech transcription results in section 3.3? If only one component is varied (cold-init model vs TWIST), then the results could be quite useful?
2. The data scaling is highlighted as a major contribution, however, except OPT other models aren't varied upto the Billion scale.

**Limitations:**

The authors have adequately addressed the limitations.

---

> ### Author Rebuttal · Authors · 2023-08-09
>
> We appreciate the reviewer’s acknowledgement of the new benchmark contributions, the benefit of using TWIST in terms of convergence speed and quality while being simple and easy to use.
>
> **Regarding reporting only one model size for Bloom/Pythia (weakness 1, question 2):** To have a fair comparison in the paper, we report results for different model sizes using the same family of model architecture, OPT. To evaluate the effect of model architecture, we additionally report results for Bloom and Pythia. We also report results for ImageGPT and LLaMA models. Hence, we believe the reported experiments are comprehensive and include a versatile set of models. We highlight that each such experiment is resource intensive (e.g., it takes ~9 days to train the 1.3B parameter model for 400k steps using 32 GPUs), which limits the number of experiments we can run. Nevertheless, we provide cold- and warm-init results for Bloom and Pythia using a 1.3B parameter models. Results show similar trends as we observe with the other reported models.
>
>
> |     TWIST     |     MODEL     |   PPL↓   |   SWUGGY↑   |   SBLIMP↑   |
> |:-------------:|:-------------:|:--------:|:-----------:|:-----------:|
> |       ✗       | BLOOM 1.3B    |   6.16   |    81.88    |    59.40    |
> |       ✓       | BLOOM 1.3B    |   6.02   |    82.87    |    60.47    |
> |       ✗       | Pythia 1.3B   |   6.13   |    81.47    |    59.72    |
> |       ✓       | Pythia 1.3B   |   5.91   |    82.73    |    60.19    |
>
> **Regarding reporting PPL over transcripts (weakness 1, question 1):**
> As stated in the paper (lines 177-182), text-PPL scores over the transcribed speech generations are  sensitive to modification in the sampling parameter and ASR errors [1]. Furthermore, text-PPL tells only one side of the story, we also need to count for diversity in the generation (such as auto-BLEU, [1]). For instance, repeated speech gets good text-PPL scores, at the cost of bad diversity measures.
>
> To mitigate that, when computing the automatic metrics for generated speech, we threshold the temperature parameter using the ASR model confidence. Next, as there can be trade-offs in terms of sentence coherence and diversity, we calibrated the temperature so that the generated speech matches the transcription of the original utterance (as much as possible) in terms of generation diversity (auto-BLEU). Due to all of the above, we decided to drop this evaluation from our manuscript. Nevertheless, as per the reviewer’s request, below we report both text-PPL and auto-BLEU results for our main family of models (OPT architecture).
>
>
> |     TWIST     |     MODEL-SIZE     |   text-PPL↓   |   auto-BLEU↓   |
> |:-------------:|:------------------:|:-------------:|:--------------:|
> |      ---      | orig. Transcript   |     36.35     |     0.225      |
> |       ✗       | 125M param         |     88.17     |     0.305      |
> |       ✓       | 125M param         |     74.95     |     0.288      |
> |       ✗       | 350M param         |    117.17     |     0.265      |
> |       ✓       | 350M param         |     92.69     |     0.253      |
> |       ✗       | 1.3B param         |     67.68     |     0.298      |
> |       ✓       | 1.3B param         |     47.8      |     0.296      |
>
> As can be seen, while the diversity metric is comparable across models (due to calibration), TWIST models outperform the cold-init. ones in terms of sentence coherence. These measures further support our claims: TWIST models outperform cold-init, and models generally improve with scale. All text-PPL measures were computed using a LLaMA-7B (text model).
> Notice, the text-PPL for the 350M param models are worse than the text-PPL of the 125M param models. However, as stated, the diversity of the generations (as computed using auto-BLEU) is better for the 350M models.
>
> **Regarding the simplicity of the proposed method:** As noted by the reviewer themselves, the simplicity of our approach is one of its strengths, especially combined with consistent performance gains that it leads to.
>
> **Regarding prior empirical justification:** To the best of our knowledge, we are the first work to empirically show this phenomenon for SpeechLMs (i.e., text initialization is beneficial for speech based LMs). Similarly, for ImageGPT, we are not aware of any prior work that shows similar findings in the past (see background and method sections). Additionally, the main purpose of the ImageGPT experiment is to better highlight the advantages of using text-based LMs for initialization. We believe such findings are interesting and would be valuable for the community. We would be happy if the reviewer could point to prior work that presents similar findings.
>
> **Regarding data scaling (Question 2):** We believe there is a misunderstanding. Scaling the data does help the model get significantly better performance across all model sizes, except for the 125M param model using 10% or 100% of the data where we observe minor improvements. Which we believe is due to model capacity (see Figure 2a). When considering model scaling, we also observe a significant improvement in performance, when scaling the model. Again, with the exception of only 1% of the data. In this case, the bigger model tends to overfit faster. Can the reviewer please clarify their question / concern?
>
> In addition, we report the results for BLOOM and Pythia using the 1.3B parameter model and observe similar trends. See the table above.
>
> [1] Lakhotia, Kushal, et al. "On generative spoken language modeling from raw audio." Transactions of the Association for Computational Linguistics 9 (2021): 1336-1354.

---

> > ### Author Response · Authors · 2023-08-19
> >
> > Dear Reviewer 6BNW,
> >
> > We would like to thank you once again for taking the time to review our manuscript! Your comments and feedback are highly appreciated.
> >
> > We would like to know whether our rebuttals addressed your previous comments. If they did, we would greatly value an increase of the score.
> > Please, let us know if you have any follow-up concerns or comments. We would be happy to clarify those.

---

### Official Review · Reviewer_qFrm · 2023-07-08

**Soundness:** 4 excellent
**Presentation:** 3 good
**Contribution:** 3 good
**Rating:** 7
**Confidence:** 4

**Summary:**

This paper studies the effect of textual LM on SpeechLMs. They propose TWIST, which initializes the SpeechLM model with a pre-trained textual LM, and then finetune with the speech datasets.
This paper provides a complementary exploration of generative spoken language modeling, including front-end processing of speech tokenizer, the core component of SpeechLM, and back-end speech synthesis of Vocoder.
TWIST uses a warm-start for SpeechLM and outperforms the cold-start model in all experiments. In addition, this paper also provides empirical results across various pretrained models, and presents the large-scale 7B-sized Speech language model.


**Strengths:**

1. This paper provides insight into the design of SpeechLMs based on rapidly developing large- scale textual pretrained models.
2. This paper presents many empirical conclusions and findings, such as the HuBERT setting, results on different scales of training data.
3. This paper introduces new benchmarks for evaluating SpeechLMs.


**Weaknesses:**

1. Some details are missing. For example, warm-start allows better performance compared to cold-start, but training costs for both methods are not reported. These results help researchers estimate the training costs of large-scale SpeechLMs.
2.  Automatic evaluation results are missed. This paper reports the human evaluation of MMOS for speech generation, but no automatic evaluation results are included, like WER for ASR results. Therefore, it is different to compare results across related work.
3. Although TWIST shows powerful capability of speech understanding, SpeechLMs still lack deep semantic understanding compared to textual models, as stated in Limitation. Therefore, the proposed method is somewhat of a compromise.

Despite these issues, I still believe that this work is highly valuable and can provide direction for the following researchers.



**Questions:**

1. Does HuBERT use different vocoders to resynthesize speech for different clustering tokens and frequencies in Section A.2?
2. Can the authors provide an explanation for the poor performance of cold-start methods? Is it because there is insufficient training data to fully train the model?


**Limitations:**

Yes

---

> ### Author Rebuttal · Authors · 2023-08-08
>
> We appreciate that the reviewer believes our work is “highly valuable and can provide direction for following research.” We also thank the reviewer for acknowledging the contributions of our paper including the rigorous empirical results, the insights into the design considerations of SpeechLMs and the introduction of a new speech benchmark based on the StoryCloze benchmark.
>
> **Regarding reporting training costs:** . We highlight that both the cold- and warm-started models are trained using similar resources and have the same computational cost, for example the 1.3B models are each trained on 32 GPUs for 400K steps (about 9 days). See Appendix A.1 for more details. Moreover, in figure 2.b we show that for some models our approach (TWIST) saves about 75% of the training time/cost compared to the cold-init baseline. Finally, we note that these numbers do not include the cost of pretraining the textual models as we assume such models are readily available, and do not require further processing on our behalf.
>
> **Regarding lack of automatic evaluations:** We agree that additional automatic evaluations would make the paper stronger. Unfortunately, automatic evaluations is a weak point of the emerging field of SpeechLMs. In fact, one of our main contributions addresses this particular aspect: the introduction of tStoryCloze and sStoryCloze as other objective metrics. In addition to these metrics, we provided the standard metrics in evaluating speechLMs:  sWUGGY and sBLIMP (see [1-3]).
>
> Furthermore, as discussed in lines 177-182, we tried to add metrics over text transcription (produced by an ASR model) of the generated speech such as sentence coherence (text-PPL) and generation diversity (auto-BLEU) as suggested in [1]. We observe that these metrics can be unstable and sensitive to different hyper-parameters. For instance, errors from the ASR model, such as word corrections for unintelligible speech, can affect the text-PPL metric. To mitigate that, we threshold the temperature parameter using the ASR model confidence. Next, as there can be trade-offs in terms of sentence coherence and diversity, e.g., repeated generations can get good text-PPL but bad diversity measures, we calibrated the temperature so that the generated speech matches the transcription of the original utterance (as much as possible) in terms of generation diversity. Due to all of the above, we decided to drop this evaluation from our manuscript. Nevertheless, as per the reviewer’s request, below we report both text-PPL and auto-BLEU results for our main family of models (OPT architecture).
>
> |     TWIST     |     MODEL-SIZE     |   text-PPL↓   |   auto-BLEU↓   |
> |:-------------:|:------------------:|:-------------:|:--------------:|
> |      ---      | orig. Transcript   |     36.35     |     0.225      |
> |       ✗       | 125M param         |     88.17     |     0.305      |
> |       ✓       | 125M param         |     74.95     |     0.288      |
> |       ✗       | 350M param         |    117.17     |     0.265      |
> |       ✓       | 350M param         |     92.69     |     0.253      |
> |       ✗       | 1.3B param         |     67.68     |     0.298      |
> |       ✓       | 1.3B param         |     47.8      |     0.296      |
>
> As can be seen, while the diversity metric is comparable across models (due to calibration), TWIST models outperform the cold-init. ones in terms of sentence coherence. These measures further support our claims: TWIST models outperform cold-init, and models generally improve with scale. All text-PPL measures were computed using a LLaMA-7B (text model).
>
> If the reviewer has additional metric suggestions, we would be happy to incorporate them into the paper.
>
> **Regarding SpeechLMs not on par with TextLMs:** We agree there is still room for improvement for speechLMs when compared to text-based models. While not perfect, we believe our work is a step in the right direction towards closing that gap, and hope it will allow others to build on our approach and make further progress.
>
> **Answers to questions:**
> 1. Yes, each HuBERT model comes along with its “own” vocoder.
> 2. We can only speculate as to why cold-initialized models perform worse than warm-initialized ones. One option would be, as the reviewer suggests, that the amount of data is still insufficient for a cold-started model. This is backed by Figure 2.a which shows that for larger models using more data makes the gap between cold- and warm-initialized models smaller. This hypothesis is not trivial to test, as we used all publicly available speech data that we are aware of, so simply scaling the models with more data is challenging, which further highlights the need for alternative methods for improving SpeechLMs, such as our proposed method.
> Having said that, the type of data used for pretraining plays an important role, as can be observed from the modality experiment, which used ImageGPT as warm initialization for TWIST (lines 237-241).
>
> [1] Lakhotia, Kushal, et al. "On generative spoken language modeling from raw audio." Transactions of the Association for Computational Linguistics 9 (2021): 1336-1354.
> [2] Borsos, Zalán, et al. "Audiolm: a language modeling approach to audio generation." IEEE/ACM Transactions on Audio, Speech, and Language Processing (2023).
> [3] Qian, Kaizhi, et al. "Contentvec: An improved self-supervised speech representation by disentangling speakers." International Conference on Machine Learning. PMLR, 2022.

---

> > ### Author Response · Authors · 2023-08-19
> >
> > Dear Reviewer qFrm,
> >
> > We would like to thank you once again for taking the time to review our manuscript! Your comments and feedback are highly appreciated.
> >
> > We would like to know whether our rebuttals addressed your previous comments. Please, let us know if you have any follow-up concerns or comments. We would be happy to clarify those.

---

> > ### Comment · Reviewer_qFrm · 2023-08-21
> >
> > Thanks for the detailed explanations. I'll keep my score, but I hope the authors can add more content as suggested by other reviewers.

---

### Author Rebuttal · Authors · 2023-08-08

We thank all reviewers for their detailed responses and valuable feedback. We are happy the reviewers acknowledge the use of our method (TWIST) benefits the training and quality of speech language models (SpeechLMs). Reviewer 6BNW mentioned that **TWIST is effective, simple and easy to use**, while reviewer 5UoA agreed that **using a warm initialization from text-LMs is a good idea**. We appreciate reviewers qFrm, 6BNW and pWSQ for acknowledging that **our work is one of the first works to perform large scale experiments on SpeechLMs, both in terms of data scale and model sizes**. We thank reviewers qFrm pWSQ and 5UoA for emphasizing that **our paper includes extensive evaluations across different textual LMs, tokenizers, models and dataset size**. Furthermore, we thank reviewer qFrm for believing that our work is **“highly valuable and can provide direction for following research”** We are grateful for reviewers qFrm, 6BNW and pWSQ noting our contribution to the speechLM community by **presenting new benchmarks (tStoryCloze and sStoryCloze)**.

A specific response for each reviewer appears in the “personal” rebuttal section.

---

> ### Author Response · Authors · 2023-08-16
>
> We would like to thank the reviewers again for taking the time to review our manuscript.
>
> As we are approaching the end of the authors-reviewers discussion period, we would be happy to address any remaining concerns per the reviewers' request.

---

### Decision · Program_Chairs · 2023-09-21

**Decision:**

Accept (poster)

**Comment:**

This paper examines the impact of pretraining with text on speech-based language models and shows how this straightforward model initialization can enhance performance, taking into account both automatic and human-generated metrics. The majority of reviewers expressed gratitude to the authors for their rebuttal, and all reviewers ultimately granted acceptance, from borderline accept to accept.